# Hyperbranched Polymers Modified with Dansyl Units and Their Cu(II) Complexes. Bioactivity Studies

**DOI:** 10.3390/ma13204574

**Published:** 2020-10-14

**Authors:** Paula Bosch, Desislava Staneva, Evgenia Vasileva-Tonkova, Petar Grozdanov, Ivanka Nikolova, Rositsa Kukeva, Radostina Stoyanova, Ivo Grabchev

**Affiliations:** 1Institute of Science and Technology of Polymers, ICTP-CSIC, Juan de la Cierva 3, 28006 Madrid, Spain; pbosch@ictp.csic.es; 2Department of textile and leather, University of Chemical Technology and Metallurgy, 1756 Sofia, Bulgaria; grabcheva@mail.bg; 3The Stephan Angeloff Institute of Microbiology, Bulgarian Academy of Sciences, 1113 Sofia, Bulgaria; evaston@yahoo.com (E.V.-T.); grozdanov_bg@yahoo.com (P.G.); vanianik@mail.bg (I.N.); 4Institute of General and Inorganic Chemistry, Bulgarian Academy of Sciences, 1113 Sofia, Bulgaria; rositsakukeva@yahoo.com (R.K.); radstoy@svr.igic.bas.bg (R.S.); 5Faculty of Medicine, Sofia University “St. Kliment Ohridski”, 1407 Sofia, Bulgaria

**Keywords:** hyperbranched polymers, dansyl, antimicrobial activity, antibacterial textile

## Abstract

Two new copper complexes of hyperbranched polymers modified with dansyl units were synthesized and characterized by infrared spectroscopy (IR) and electron paramagnetic resonance (EPR) techniques. It was found that copper ions coordinate predominantly with nitrogen or oxygen atoms of the polymer molecule. The place of the formation of complexes and the number of copper ions involved depend on the chemical structure of the polymer. The antimicrobial activity of the new polymers and their Cu(II) complexes was tested against Gram-negative and Gram-positive bacterial and fungal strains. Copper complexes were found to have activity better than that of the corresponding ligands. The deposition of the modified branched polymers onto cotton fabrics prevents the formation of bacterial biofilms, which indicates that the studied polymers can find application in antibacterial textiles.

## 1. Introduction

In recent years, antimicrobial resistance has become an alarming threat to public health. Different pathogens have already become resistant to clinically used antibiotics [1,2,3]. A large number of viruses, fungal and protozoan pathogens are becoming progressively resistant to the drug therapies used currently. This, in turn, greatly reduces the choice of alternative treatment. Unfortunately, in many cases, this is impossible; therefore, the search for new effective compounds against pathogens causing common infections has been crucial. In addition, the need to use effective antibacterial and antifungal materials in everyday life, such as in hospital and community settings and on public transport, among others, is constantly increasing [4].

One possibility of finding novel compounds with well-expressed antimicrobial activity to implement dendrimers or hyperbranched polymers with a large number of functional groups that can be functionalized with biologically active substances [5,6,7,8]. On the other hand, the use of some biologically important metal ions, incorporated into such structures, enhances the effect of the active substances against pathogenic microorganisms [9,10]. Recently, dendrimers modified with 1,8-naphthalimide poly(propylene imines) (PPI) or poly(amidoamine) (PAMAM) and their Cu(II) or Zn(II) complexes have been investigated against different pathogens in solution and after their deposition onto textile materials [11,12,13,14,15]. Hyperbranched polymers modified with 1,8-naphthalimides [16] or acridine [17] units have shown very good sensor properties in detecting metal ions and protons. These polymers also exhibit promising antimicrobial activity [18,19]. Recently, in order to broaden the selection of fluorophores in the modification of polymers with potential bioactivity, hyperbranched polymers containing dansyl groups were synthesized and their photophysical characteristics were investigated [20]. The incorporation of dansyl fluorophores into the structure of substances with antibiotic activity leads to the production of fluorescent antibiotics, which can provide knowledge on the mechanism of their interaction with resistant bacteria [4].

The aim of this work was to investigate the bioactivity of two different hyperbranched polymers modified with dansyl units and their Cu(II) complexes against different pathogens in solution and after their deposition onto cotton fabric with regards to obtaining antibacterial textile materials.

## 2. Materials and Methods

### 2.1. Materials

The synthesis, spectral and photophysical characteristics of hyperbranched polymers S1 and S2 presented in Scheme 1 were described recently [20]. The same polymers were used to prepare the copper complexes in this work. Dimethyl sulfoxide (DMSO) for molecular biology and Cu(NO_3_)_2_·3H_2_O were used as obtained from Sigma-Aldrich (Darmstadt, Germany). The used methods for characterization and microbiological activity of polymers are presented in the Appendix A.

### 2.2. Synthesis of the Cu(II) Complex with S1

Polymer S1 (0.306 g, 10 mmol) was dissolved in 10 mL of ethanol and Cu(NO_3_)_2_·3H_2_O (0.120 g, 50 mmol) was added. The mixture was stirred for 2 h and the solid complex formed was filtered off, washed with ethanol (5 × 10 mL) and dried. Yield: 0.275 g, 86.9%;

FTIR (cm^−1^): 1720, 1711, 1638, 1574, 1459, 1306, 1263, 1140,1075, 789, 624.

Analysis: C_152_H_167_N_35_O_24_S_6_Cu_3_ (3247.92 g mol^−1^): Calc. (%): C-56.16, H- 5.14, N- 15.08. Found (%): C-56.22, H- 5.08, N-15.29;

### 2.3. Synthesis of the Cu(II) Complex with S2

A mixture of polymer S2 (0.650 g, 10 mmol) dissolved in 10 mL of ethanol and Cu(NO_3_)_2_·3H_2_O (0.290 g, 120 mmol) was prepared and was stirred for 2 h. The solid complex formed was filtered off, washed with ethanol (5 × 10 mL) and dried. Yield: 0.598 g, 82.0%;

FTIR (cm^−1^): 1731, 1656, 1613, 1573, 1389, 1308, 1230, 942, 789, 619.

Analysis: C_301_H_408_N_60_O_96_S_12_Cu_8_ (7289.22 g mol^−1^): Calc. (%): C-49.55, H- 5.60, N-11.52. Found (%): C-49.64, H-5.68, N-11.39;

### 2.4. Treatment of Cotton Fabric with S1 and S2

Polymers S1 or S2 (5.0 mg) and their Cu(II) complexes were dissolved in 5 mL *N,N*-dimethylformamide. The cotton fabric (1 g, weight 140 g/m^2^) was immersed into the solution at 40 °C for 60 min. Then, the cotton fabric was washed with water and dried at ambient temperature. The coloristic characteristics (*L*, a* b*, XYZ, xy* and whiteness) of the treated cotton fabrics were determined by the Datacolor colorimetric system.

## 3. Results

### 3.1. Chemical Structure of S1 and S2 and Their Cu(II) Complexes 

Figure 1 shows the change in fluorescence intensity during the titration of S1 and S2 with copper ions. As seen, in both cases, quenching of the fluorescence intensity of polymers was observed. When using S1 as a ligand, three copper ions formed a complex with one polymer molecule, while in the case of polymer S2, the complex was formed by one polymer ligand with eight copper ions.

To clarify the structure of the copper complexes, IR spectral and EPR analysis of S1 and S2 ligands and the respective copper complexes [Cu_3_(S1)] and [Cu_8_(S2)] were performed. Figure 2 shows the chemical structure of the copper complex of S1 and its IR spectrum, which is compared to that of the initial ligand. Data suggest that the copper ions form a coordinative bond with nitrogen atoms. As Figure 2A demonstrates, the copper ions deprotonate the secondary amino group of the sulfonamide group and the signal at 1609 cm^−1^ found in the S1 spectrum disappears in that of the copper complex. This probably changes the polarization of S=O from the sulfonamide structure and the intensity of the characteristic vibration peak at 1357 cm^−1^ decreases significantly [21]. A change in the intensity of the peak was also observed at 1660 cm^−1^ due to the vibration of the amide group after the complex had been formed. That gives a reason to propose the formation of a coordinate bond between the copper ions and the nitrogen atoms as shown in Figure 2A. On the other hand, the characteristic peak for S=0 at 1142 cm^−1^ does not change its position and intensity, indicating that the oxygen atom does not coordinate with the copper ions [22]. These observations were confirmed by the solid-state EPR analysis of the compounds. Similar spectral behavior was observed for S2 and [Cu_8_(S2)].

The EPR spectrum of the complex of [Cu_3_(S1)] has an anisotropic signal with the *g*-components of *g*_1_ = 2.268, *g*_2_ = 2.065 and *g_3_* = 2.076 (Figure 3A). The hyperfine structure is resolved only in the region of the g_1_-component, the hyperfine constant being A_1_ = 16.5 mT. Close inspection of the EPR parameters reveals that the complex of Cu (II) with S1 exhibits a relatively low value of the g_1_-component and a high magnitude of the hyperfine constant. The observed relation between the g_1_ and A_1_ values implies that three Cu(II) ions are mainly coordinated by the nitrogen atoms in the ligand [23,24]. The EPR spectrum of the complex of [Cu_8_(S2)] is shown in Figure 3B. The spectrum displays an anisotropic signal with the *g*-components of *g*_1_ = 2.304, *g*_2_ = 2.085 and *g*_3_ = 2.047. The *g*-values remain unchanged during cooling from 295 to 120 K. In the range of the g_1_-component, a hyperfine structure with a constant of A_1_ = 14.5 mT is observed. The relatively high value of the *g_1_*-component, together with the low magnitude of the hyperfine constant A_1_, is an indication of the preferential coordination of Cu(II) ions through oxygen atoms [23]. In this case, probably due to the larger number of oxygen coordination centers, copper ions prefer to coordinate with them instead of with the nitrogen atoms, as in the case of [Cu_3_(S1)]. The quantitative EPR analysis shows that around eight Cu(II) ions are bound to the S2 ligand.

### 3.2. Deposition of Hyperbranched Polymers onto the Cotton Fabric 

In order to obtain antibacterial cotton fabrics, cotton fabric weighing 140 g/m^2^ was loaded with the tested polymers by the irrigation method for 60 min.

Fluorescence spectroscopy was used to determine the number of hyperbranched polymers deposited onto the cotton surface. It was found that 4.2 mg of S1 (88%), 4.4 mg of [Cu_3_(S1)], 4.0 mg of S2 (81%) and 4.3 mg of [Cu_8_(S2)] were loaded onto the cotton surface. These results show very good extraction and retention of the studied polymers on the cotton surface. Due to the fact that the tested compounds have a high molecular weight and are insoluble in water, they adhere firmly to the cotton matrix. The retention of macromolecular compounds itself is due to the formation of a large number of hydrogen bonds and weak van der Waals interactions.

The description of the photophysical characteristics of the hyperbranched polymeric ligands S1 and S2 revealed their significant solvatochromism. Both dansylsulfonamide derivatives absorb in the UV region at 332–340 nm and have fluorescence maxima at 454–528 nm with a wide Stokes shift [20]. This property is preserved after their deposition onto the cotton fabric. Figure 4 shows the excitation and fluorescence spectra of cotton fabric treated with S1 and [Cu_3_(S1)] with maxima at 340 nm and 506 nm. Obviously, the intensity of the emitted fluorescence decreases about three times after the formation of the copper complex. Fluorescence quenching can be explained by a decrease in the polarization of the chromophore system. Similar properties were observed in the cases of fabrics treated with S2 and [Cu_8_(S2)].

The Datacolor system was used to determine the color parameters of the cotton materials, using the parameters *L*a*b*, XYZ* and *xy*. The results are summarized in Table 1. The virgin cotton fabric, which was white in color, served as a control. Due to the fact that the studied polymers absorb in the UV region, these compounds are colorless and do not significantly change the original color of the cotton fabric. On the other hand, they reduce the degree of its whiteness. This is due to the color of the emitted fluorescence, which has a yellowish tinge. Copper complexes further reduce the whiteness of the fabric (Table 1). That indicates that those compounds do not exhibit the property of optical brighteners.

### 3.3. Biological Properties 

#### 3.3.1. Antimicrobial Activity and MIC

For the antimicrobial tests, the strains *P. aeruginosa*, *B. cereus* and *C. lipolytica* were selected as opportunistic pathogens representative of Gram-negative bacteria, Gram-positive bacteria and fungi. The antimicrobial activities of the compounds were assessed by determining/observing the presence of inhibition zones and MIC values. Applied at concentrations of 46.55 nM for [Cu_3_(S1)] and 3.14 nM for [Cu_8_(S2)], the compounds show good activity against the tested strains. For some of the strains, ligand S1 demonstrates slightly higher activity than S2, and the Cu (II) complexes turned out to be slightly more active than the ligands. The observed inhibition zones are in the range of 11–12 mm for the ligands and 12–14 mm for their respective complexes (Table 2).

As seen from Table 3, the compounds have MIC values in the range of 16.46 ÷ 93.11 µmol/L. A lower MIC value indicates that a smaller amount of the sample is required for inhibiting the growth of the organism; therefore, drugs with lower MIC scores are more effective antimicrobial agents. The compounds are effective in inhibiting the growth of the Gram-positive *B. cereus* strain followed by that of the yeasts *C. lipolytica.* The highest MICs were determined towards *P. aeruginosa*. The level of activity of the ligands and their Cu(II) complexes was considerably lower than that of the reference drugs gentamicin and nystatin. A comparison of the antimicrobial activity of polymers shows that dendrimer S2 exhibits approximately double the activity against the pathogens tested compared to that of S1. On the other hand, in both cases, the copper complexes [Cu_3_(S1)] and [Cu_8_(S2)] showed increased antimicrobial activity in comparison to ligands S1 and S2. 

#### 3.3.2. Antimicrobial Activity of the Treated Cotton Fabrics

The antimicrobial effect of cotton fabrics treated with S1 and S2 compounds and their Cu-complexes was evaluated by the reduction of the growth of *B. cereus* and *P. aeruginosa*. Since the tested compounds have a high molecular weight and are insoluble in water, they adhere firmly to the cotton matrix. The direct contact with bacterial cells contributes to the antimicrobial effect of the treated cotton fabrics in solution. We found a low reduction of the growth of the test strains by the cotton samples treated with ligands S1 and S2. A slightly enhanced antibacterial activity of the cotton samples treated with metal complexes was observed (Figure 5).

These results clearly show that the tested compounds are firmly attached to the surface of the cotton fabric and the probability of migrating from it is minimal. This, in turn, means that they prevent the formation of a bacterial biofilm on the cotton surface. The effect of the cotton fabric samples treated with the polymers on the adhesion and biofilm formation of pathogen strains was investigated by SEM. Figure 6 shows SEM images of the virgin cotton fabrics and those treated with S1 and [Cu3 (S1)] at 10,000× magnification after their contact with *B. cereus.*
Figure 6A shows the typical structure of the cotton fabric before its treatment with the polymers. A colony of bacterial cells and biofilm formed on the cotton surface is shown in Figure 6B. A significant reduction of the adhesion and biofilm formation is observed on the cotton fabric treated with S1 (Figure 6C) and only single cells are attached to the cotton surface treated with the [Cu_3_(S1)] complex (Figure 6D). These results demonstrate that the copper complexes of S1 and S2 inhibit the attachment of bacteria to the surface of the cotton fabric thus preventing the formation of bacterial biofilm. This reveals the suitability of the new polymers to be used for the preparation of antibacterial textiles.

#### 3.3.3. Cytotoxicity

The toxicity of the substances was investigated towards a model HEp-2 cell line. The dose–response curves are presented in Figure 7 and Figure 8. The ligands S1 and S2 showed similar toxicity with CC_50_ values of 202.7 μg/mL and 208.7 μg/mL, respectively. The Cu-complexes demonstrated slightly lower activity, compared to that of the ligands, and [Cu_8_(S2)] was found to be slightly less toxic than [Cu_3_(S1)] with CC_50_ values of 234.2 μg/mL and 219.1 μg/mL, respectively. Polymer S1 is six times less toxic in comparison with the same polymer modified with 1,8-naphthalimide [18] and twice less toxic than the polymer modified with acridine [19].

## 4. Conclusions

The chemical structures of two hyperbranched polymers and their copper complexes were identified and proven by IR and fluorescence spectroscopy and EPR. It was found that copper ions coordinate with nitrogen or oxygen atoms depending on the structure of polymer ligands. The number of coordinated copper ions also depends on the polymer structure. The antimicrobial activity of the copper complexes against various pathogenic microorganisms was studied and compared with that of their ligands. The new polymers demonstrated better activity against Gram-positive bacteria compared to Gram-negative bacteria. On the other hand, the copper complexes have a more pronounced activity than the respective ligands. This trend continued after the new compounds were deposited onto cotton fabric. SEM studies showed that copper complexes prevent the retention of bacteria on the cotton surface and the formation of a bacterial biofilm, which indicates that the compounds are suitable for use in the preparation of antibacterial textiles. The cytotoxicity studies of the copper complexes and polymer ligands demonstrated that copper ions reduce the cytotoxicity of the modified hyperbranched polymers, thus expanding the prospects of their biomedical applications.

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
