# Peer review of "Hyperbranched Polymers Modified with Dansyl Units and Their Cu(II) Complexes. Bioactivity Studies"

_materials, 2020, doi:10.3390/ma13204574_

Round 1
Reviewer 1 Report
Antibacterial/ antimicrobial textile products have enjoyed great interest recently because of their potential in restricting spreading of infections. The paper submitted for review concerns this area of research. It is interesting but in my opinion prior to acceptance for publication, the authors need to address a few points.
- The Introduction includes 21 references of which 13 are autocitations, which is disproportionate.
- Lines 61 and 68
According to the authors, the complexes of Cu(II) with hyperbranched polymers S1 or S2 were synthesized in the same amounts and with the same yield of 0.275g and 86.9%, respectively. Please explain how it was possible? Is it just a mistake?
- Lines 63-64 and 70-71
No name and producer of the apparatus for elemental analysis and no error of measurements are given. Please supplement the information.
- Lines 137-139
The sentences “The Cu(II) complexes of hyperbranched polymer ligands S1 and S2 were prepared by reacting their solution in ethanol with copper nitrate at 40oC for 2 hours. The precipitate formed was filtered off and washed with ethanol.”, should be moved to the Experimental section, the more so that some of the information given in them (except for temperature) have been repeated (lines 59-61 and lines 66-68).
- The number of copper(II) ions making complexes with polymer ligands were spectrophotometrically determined by fluorescence spectroscopy (Figure 1). Moreover, Figure 4 presents the excitation (Ex) and fluorescence (Em) spectra of a cotton fabric treated with S1 and its [Cu3(S1)] complex. In the experimental section there is no mention on these measurements likewise no mention on the IR measurements – lines 148-158 and Figure 2b.
- 2 presents the chemical structure of [Cu3(S1)] complex and its FTIR spectrum compared with that of a free ligand S1. The authors should present the analogous data for the complex [Cu8(S2)] the more so that Fig.3 presents the EPR spectra of these two compounds.
- Lines 163-176
The authors should give some references supporting the conclusions of the coordination through nitrogen or oxygen atoms.
- Lines 211-212
Why the difference in the concentrations of the compounds used is so great, 46.55nM for [Cu3(S1)] and 3.14nM for [Cu8(S2)]?
- Many antimicrobial agents are gradually removed from the textiles which reduces the antimicrobial activity with time. The majority of the hitherto proposed agents is to some degree water soluble and the antibacterial/ antimicrobial effects decrease with time. According to the authors of the paper submitted, the compounds they propose [Cu3(S1)] and [Cu8(S2)] are insoluble in water (as described in lines 185-187). However, I wonder if the authors have checked the effect of a multiple washings and evidenced the permanent antibacterial effect.
- The first two sentences of the Conclusion are redundant “Copper complexes…….and EPR” , lines 268-270. Moreover, the form of the Conclusion is too general and looks more like an abstract, I would suggest rewriting it.
Author Response
Thanks to the reviewer for the constructive remarks that improved the quality of our manuscript.
Antibacterial/ antimicrobial textile products have enjoyed great interest recently because of their potential in restricting spreading of infections. The paper submitted for review concerns this area of research. It is interesting but in my opinion prior to acceptance for publication, the authors need to address a few points.
- The Introduction includes 21 references of which 13 are autocitations, which is disproportionate.
The autocitations have been reduced
- Lines 61 and 68
According to the authors, the complexes of Cu(II) with hyperbranched polymers S1 or S2 were synthesized in the same amounts and with the same yield of 0.275g and 86.9%, respectively. Please explain how it was possible? Is it just a mistake?
It was corrected.
- Lines 63-64 and 70-71
No name and producer of the apparatus for elemental analysis and no error of measurements are given. Please supplement the information.
Done
- Lines 137-139
The sentences “The Cu(II) complexes of hyperbranched polymer ligands S1 and S2 were prepared by reacting their solution in ethanol with copper nitrate at 40oC for 2 hours. The precipitate formed was filtered off and washed with ethanol.”, should be moved to the Experimental section, the more so that some of the information given in them (except for temperature) have been repeated (lines 59-61 and lines 66-68).
It has been removed from the text
- The number of copper(II) ions making complexes with polymer ligands were spectrophotometrically determined by fluorescence spectroscopy (Figure 1). Moreover, Figure 4 presents the excitation (Ex) and fluorescence (Em) spectra of a cotton fabric treated with S1 and its [Cu3(S1)] complex. In the experimental section there is no mention on these measurements likewise no mention on the IR measurements – lines 148-158 and Figure 2b.
2 presents the chemical structure of [Cu3(S1)] complex and its FTIR spectrum compared with that of a free ligand S1. The authors should present the analogous data for the complex [Cu8(S2)] the more so that Fig.3 presents the EPR spectra of these two compounds.
The equipment used was added
- Lines 163-176
The authors should give some references supporting the conclusions of the coordination through nitrogen or oxygen atoms.
The references were added
- Lines 211-212
Why the difference in the concentrations of the compounds used is so great, 46.55nM for [Cu3(S1)] and 3.14nM for [Cu8(S2)]?
In this case, the amount of polymers depends on their molecular weight, which is higher at [Cu8(S2)]
- Many antimicrobial agents are gradually removed from the textiles which reduces the antimicrobial activity with time. The majority of the hitherto proposed agents is to some degree water soluble and the antibacterial/ antimicrobial effects decrease with time. According to the authors of the paper submitted, the compounds they propose [Cu3(S1)] and [Cu8(S2)] are insoluble in water (as described in lines 185-187). However, I wonder if the authors have checked the effect of a multiple washings and evidenced the permanent antibacterial effect.
Some of the antibacterial textile in the clinical practice is used once, such as wound dressings, gauzes and others. At this stage we have not checked the effect of repeated washings of our textile.
- The first two sentences of the Conclusion are redundant “Copper complexes…….and EPR” , lines 268-270. Moreover, the form of the Conclusion is too general and looks more like an abstract, I would suggest rewriting it.
The conclusion has partially been revised.
Reviewer 2 Report
The methods and presentation of the biological part have to be improved:
1) the choice of bacterial and fungal strains should be justified. Why Authors choose just these 3 rods? Explain their importance. Add more strains, e.g. Staplycococci.
2) Table 2 - the letter G and Ns have to be explained in the legend
3) the MIC values of both standards (Gentamycin and Nystatin) should be obligatory added to the Table 3. Adding them, we just know, what is the quantity of antimicrobial potential of ligands and copper complexes. The level of activities of S1/S2 contra their complexes is the same. I suppose it is considerably lower than that of reference drugs.
4) cytotoxicity was performed only using one cancer cells. The toxicity of new derivatives towards normal cell lines (e.g. HaCaT) have to be added, and compared to pathological HEp-2 cells.
5) after adding results suggested in points 3) and 4), discussion should be rewritten.
6) language editing should be done.
Author Response
the methods and presentation of the biological part have to be improved:
1) the choice of bacterial and fungal strains should be justified. Why Authors choose just these 3 rods? Explain their importance. Add more strains, e.g. Staphylococci.
The strains P. aeruginosa, B. cereus and C. lipolytica were selected as opportunistic pathogens representatives of Gram-negative and Gram-positive bacteria and fungi.
2) Table 2 - the letter G and Ns have to be explained in the legend
Was done
3) the MIC values of both standards (Gentamycin and Nystatin) should be obligatory added to the Table 3. Adding them, we just know, what is the quantity of antimicrobial potential of ligands and copper complexes. The level of activities of S1/S2 contra their complexes is the same. I suppose it is considerably lower than that of reference drugs.
MIC values of Gentamycin and Nystatin are added in Table 3.
4) cytotoxicity was performed only using one cancer cells. The toxicity of new derivatives towards normal cell lines (e.g. HaCaT) have to be added, and compared to pathological HEp-2 cells.
The neutral red uptake assay measures cell viability measuring the accumulated in lysosomes of living cells. We have many different types of cell lines in our cell culture collection but at the moment we do not have normal cell line derived from human normal tissue. That is why we conducted the experiment with more appropriate cancer cell line with human origin.
5) after adding results suggested in points 3) and 4), discussion should be rewritten.
Some explanation has been done
6) language editing should be done.
The English has been checked
Reviewer 3 Report
This paper describes the synthesis and characterization of new copper complexes with hyperbranched polymers.
The paper is well documented and provides new experimental data that sustain the purpose of work.
However, there are some corrections to do:
There are some abbreviations without explanation:
- page 3, row 80 MPB;
- page 4, row 130 PBS.
In Materials and methods section, the complex Cu-(S1) contains 6 copper atoms (page 2, row 63) while in entire manuscript is formulated Cu3(S1).
The abbreviation for liter is L not l:
- page 3, row 96, mL instead of ml;
- page 3, row 104 mg/mL instead of mg/ml;
- page 4, row 123, mg/mL instead of mg/ml;
- page 7, row 218, mmol/L instead of mmol/l;
- page 7, table 3, mmol/L instead of mmol/l;
- page 9, rows 260 and 262, mg/mL instead of mg/ml;
Author Response
The paper is well documented and provides new experimental data that sustain the purpose of work.
However, there are some corrections to do:
There are some abbreviations without explanation:
- page 3, row 80 MPB;
- page 4, row 130 PBS
Done in the Experimental part
In Materials and methods section, the complex Cu-(S1) contains 6 copper atoms (page 2, row 63) while in entire manuscript is formulated Cu3(S1).
The correct one is 3 copper ions.
The abbreviation for liter is L not l:
- page 3, row 96, mL instead of ml;
- page 3, row 104 mg/mL instead of mg/ml;
- page 4, row 123, mg/mL instead of mg/ml;
- page 7, row 218, mmol/L instead of mmol/l;
- page 7, table 3, mmol/L instead of mmol/l;
- page 9, rows 260 and 262, mg/mL instead of mg/ml;
Corrected
Reviewer 4 Report
The manuscript by Bosch et al. describes the synthesis, characterization, and antimicrobial activity and of dansyl modified hyperbranched polymers. The report describes exciting and significant results and procedures. The major weakness of the paper is the scarcity of characterizations. Under normal circumstances, I would ask for further analyses. Given the global situation, I ask for additional explanations and softening of the text to convey that the characterizations are tentative rather than definitive.
- The manuscript implies that spectral and photophysical characterizations of the hyperbranched polymers have been performed previously. Would it be possible to include some of the characterizations of the polymers used in this study to verify that these newly synthesized samples are close to the previously synthesized ones?
- The text describes the IR data as definitive structural characterizations. IR peak assignments are far from certain. I ask that the authors include references to the IR spectra of similar compounds to justify their peak assignments and conclusions. I also ask to change the language from “it is seen” to the “data suggests.”
- Same to be said about the conclusions drawn from the EPR spectra. I ask that the author include references to EPR analyses of similar compounds to justify their interpretations.
- Is it possible to compare the antimicrobial properties of the hyperbranched polymers with commonly used antimicrobial agents?
Author Response
Thanks to the reviewer for the constructive remarks that improved the quality of our manuscript.
- The manuscript implies that spectral and photophysical characterizations of the hyperbranched polymers have been performed previously. Would it be possible to include some of the characterizations of the polymers used in this study to verify that these newly synthesized samples are close to the previously synthesized ones?
Done in the text
- The text describes the IR data as definitive structural characterizations. IR peak assignments are far from certain. I ask that the authors include references to the IR spectra of similar compounds to justify their peak assignments and conclusions. I also ask to change the language from “it is seen” to the “data suggests.”
Done
- Same to be said about the conclusions drawn from the EPR spectra. I ask that the author include references to EPR analyses of similar compounds to justify their interpretations.
Done
- Is it possible to compare the antimicrobial properties of the hyperbranched polymers with commonly used antimicrobial agents?
Antimicrobial properties of polymers have been compared to the used in the clinical practice gentamicin and nystatin
Round 2
Reviewer 2 Report
The article can be published in the presented form.
Author Response
Thank you to the reviewer for the consideration of our work and the positive evaluation.
Reviewer 4 Report
The authors sure don't make it easy to accept the manuscript. Most of the responses to my comments are "done" without any details. In my experience submitting to MDPI journals, the instructions specifically say that these brief responses to reviewers are insufficient. I ask that the authors and the editors ensure that the authors' responses are compliant and detailed.
The authors so far have not sufficiently addressed some of my comments:
1) I asked to include characterizations of S1 and S2 polymers before complexation. The authors claim that the characterizations have been done in a previous publication. However, it is unclear that the freshly prepared polymers are the same as the previously reported ones. Unless the batch of polymers used here is the same as in reference 20, some sort of validation needs to be included. The issue is not yet resolved.
2) I have asked to add further references and justification for the IR and EPR interpretations. The authors added a couple more references to interpret the IR and the EPR, but since no specific detail is given in the response, it is hard to see the difference between the 2 manuscript version. Throwing a couple of references is insufficient. Specific statements need to be made to compare the previous results to the current interpretations.
There is a mistake in reference 24.
Author Response
We thank the reviewer for the remarks and have tried to comment on them.
I asked to include characterizations of S1 and S2 polymers before complexation. The authors claim that the characterizations have been done in a previous publication. However, it is unclear that the freshly prepared polymers are the same as the previously reported ones. Unless the batch of polymers used here is the same as in reference 20, some sort of validation needs to be included. The issue is not yet resolved.
For the preparation of the copper complexes the same polymers have been used which have described and published in literature 20 (page 2 rows 54 and 55). This means that they have not been re-synthesized using the one described in ref. 20 method. For this reason, we have not considered it necessary to give their characterizations again. Some additional clarification was made in the text on page 2 of line 55 and 56.
2) I have asked to add further references and justification for the IR and EPR interpretations. The authors added a couple more references to interpret the IR and the EPR, but since no specific detail is given in the response, it is hard to see the difference between the 2 manuscript version. Throwing a couple of references is insufficient. Specific statements need to be made to compare the previous results to the current interpretations.
Several additional references have been added examining the use of IR and EPR methods to innervate the results obtained. The text describes the main characteristic bands in the IR region and their change after the formation of the copper complex. Unfortunately, we could not compare these results with similar ones, because this modified polymer and its copper complex were obtained for the first time and we do not find a basis for comparison. This method was used as indirect evidence for the formation of the copper complex based on the change in the position and intensity of some characteristic bands. Also, due to the paramagnetic nature of copper ions, the cooper complex cannot be examined by NMR spectroscopy like the polymer ligand. EPR has been used and on the basis of the obtained data for the respective constants we make a conclusion with which elements Cu(II) ions form a coordinate connection. Here also we do not see possibility of comparisons to be made with previous research. In this section some new literature sources have also been added concerning the formation of complexes with complex literature sources, but we give an interpellation to our compounds.
There is a mistake in reference 24.
The references 24 has been corrected.